# Methionine and cysteine oxidation are regulated in a dose dependent manner by dietary Cys intake in neonatal piglets receiving enteral nutrition

Anna K. Shoveller[1]¤*, Julia G. Pezzali[2], James D. House[3], Robert F. Bertolo[4], Paul B. Pencharz[1,5,6], Ronald O. Ball[1,5,7]

1 Department of Agricultural, Food and Nutritional Science, University of Alberta, Edmonton, Alberta, Canada, 2 Department of Animal Biosciences, University of Guelph, Guelph, Ontario, Canada, 3 Department of Food and Human Nutritional Sciences, University of Manitoba, Winnipeg, Manitoba, Canada, 4 Department of Biochemistry, Memorial University of Newfoundland, St. John's, Newfoundland, Canada, 5 The Research Institute, The Hospital for Sick Children, Toronto, Canada, 6 Department of Paediatrics, University of Toronto, Toronto, Ontario, Canada, 7 Department of Nutritional Sciences, University of Toronto, Toronto, Ontario, Canada

¤ Current address: Department of Animal Biosciences, University of Guelph, Guelph, Ontario, Canada
* ashovell@uoguelph.ca

**Data Availability Statement:** All relevant data are within the manuscript and its Supporting information files.

## Abstract

Methionine (Met) is an indispensable amino acid (AA) in piglets. Met can synthesize cysteine (Cys), and Cys has the ability to reduce the Met requirement by 40% in piglets. However, whether this sparing effect on Met is facilitated by downregulation of Cys synthesis has not been shown. This study investigated the effects of graded levels of Cys on Met and Cys oxidation, and on plasma AA concentrations. Piglets (n = 32) received a complete elemental diet via gastric catheters prior to being randomly assigned to one of the eight dietary Cys levels (0, 0.05, 0.1, 0.15, 0.2, 0.25, 0.40, 0.50 g kg$^{-1}$d$^{-1}$) with an adequate Met concentration (0.25g kg$^{-1}$d$^{-1}$). Constant infusion of L-[1-$^{14}$C]-Met and L-[1-$^{14}$C]-Cys were performed for 6 h on d 6 and d 8 to determine Met and Cys oxidation, respectively. Met oxidation decreased as Cys intake increased (P<0.05). At higher Cys intakes (0.15 to 0.5g kg$^{-1}$d$^{-1}$), Met oxidation decreased (P<0.05) at a slower rate. Cys oxidation was similar (P>0.05) among dietary Cys intakes; however, a significant polynomial relationship was observed between Cys oxidation and intake (P<0.05, R$^2$ = 0.12). Plasma Met concentrations increased (P<0.05) linearly with increasing levels of dietary Cys, while plasma Cys concentrations changed (P<0.05) in a cubic manner and the highest concentrations occurred at the highest intake levels. Increasing dietary levels of Cys resulted in a reduction in Met oxidation until the requirement for the total sulfur AA was met, indicating the sparing capacity by Cys of Met occurs through inhibition of the transsulfuration pathway in neonatal piglets.

**Funding:** This work was supported by grants from the Alberta Pork, Alberta Agricultural Research Institute, (Canadian Institutes of Health Research Fund # 12928) and the Natural Sciences and Engineering Research Council of Canada. (JDH) The funders had no role in study design, data collection and analysis, decision to publish, or preparation of the manuscript. Alberta Pork: https://www.albertapork.com/ Canadian Institutes of Health Research:https://cihr-irsc.gc.ca/e/193.html Natural Sciences and Engineering Research Council of Canada: https://www.nserc-crsng.gc.ca/Index_eng.asp.

**Competing interests:** The authors have declared that no competing interests exist.

## Introduction

Methionine (Met) is an indispensable amino acid (AA) necessary for protein synthesis and normal growth in mammals. Furthermore, Met plays unique roles in metabolism, as it serves as the primary methyl donor in the body, via the transmethylation pathway [1], and as the substrate for cysteine (Cys) synthesis through the transsulfuration pathway [2]. Briefly, Met is converted to S-adenosylmethionine (SAM) which can donate its methyl group to a variety of acceptors. After transferring its methyl group, SAM is converted to S-adenosylhomocysteine and homocysteine (Hcy), which represents a critical branch point as it can be remethylated to form Met or irreversibly catabolized to Cys via the transsulfuration pathway. In the latter pathway, Hcy condenses with serine, via cystathionine-β-synthase (CBS, EC 4.2.1.22), to produce cystathionine which is then hydrolyzed to α-ketobutyrate and Cys via cystathionine γ-lyase (CGL; EC 4.4.1.1).

Having a deeper understanding of the metabolism and requirements of sulfur amino acids (SAA) has several implications on mammalian health. For example, hyperhomocysteinemia has been recognized as an independent risk factor for several pathologies such as neurological and cardiovascular diseases [3] in adults and neonates. With this regard, the piglet model has been shown to serve as an appropriate model of the human neonate to investigate physiological and pathological conditions. We carried out a series of indicator AA oxidation studies to determine the requirement of sulfur amino acids in neonatal piglets. We estimated a mean total SAA (TSAA) requirement of 0.42 g kg$^{-1}$ d$^{-1}$ for enterally fed piglets [4]. Subsequently, we estimated the requirement for Met in the presence of excess Cys (0.5 g kg$^{-1}$ d$^{-1}$) and found a mean Met requirement of 0.25 g kg$^{-1}$ d$^{-1}$ for enterally fed piglets [5]. The TSAA requirement (0.42 g kg$^{-1}$ d$^{-1}$) and the capacity of Cys to spare the Met requirement (40% of the Met requirement), closely compare to the recommendation of the National Research Council [6] and with previous estimates of the SAA requirement. However, a series of experiments examining the response of Met kinetics to varying intakes of Met and Cys suggested that there was no Cys sparing mechanism in humans [7–12]. These studies provided the sulfur AAs at a level consistent with the 1985 FAO/WHO/UNU, which were identified as being inadequate [13,14]. This may explain the absence of the Cys sparing effect on Met in those studies. In previous research [15], feeding Cys resulted in the reduction of CBS activity, providing a mechanism by which Cys exerts its sparing effect upon the Met requirement. Others have used growth, feed efficiency, survival rate, apparent digestibility of Met and/or nitrogen balance to demonstrate that Cys can replace part of the Met requirement [16–22]. Furthermore, the dietary Met:Cys ratio regulates the transmethylation, remethylation and transsulfuration and that the inclusion of dietary cysteine reduces transsulfuration indicating a Cys-sparing effect [23]. More recently, Chen et al. [24] reported that the Met cycle was regulated in mice fed low-protein diets by the Met:Cys ratio via modulation of gene expression of key enzymes, such as betaine-homocysteine S-methyltransferase. To our knowledge, there are no published studies investigating the effect of graded levels of Cys on methionine oxidation (transsulfuration), as the main outcome to determine the Cys-sparing effect, on Cys oxidation, and on plasma concentrations. Thus, the present study was designed to directly determine whether increasing Cys intake, when Met intake is held constant at 50% of the recommended TSAA requirement [6], results in a change in transsulfuration (as measured by Met oxidation), Cys oxidation and plasma amino acid concentrations. As the oxidation of α-ketobutyrate (from CGL activity) releases the 1-carbon of Met, the measurement of Met oxidation provides a measurement of Cys synthesis or transsulfuration. If dietary Cys reduces Cys synthesis via a reduction in transsulfuration, then transsulfuration (as represented by Met oxidation) will be reduced as dietary Cys is increased. We hypothesized that Met oxidation will be lower at higher intakes of Cys and that Cys oxidation will remain low until the requirement for the TSAA has been met and then increase in a linear fashion.

## Material and methods

### Piglets and study protocol

The Faculty of Agriculture, Forestry and Home Economics Animal Policy and Welfare Committee at the University of Alberta approved all procedures in this study which was conducted in 2002–2003. A total of 32 male Landrace/Large White intact piglets (Genex Swine Group) were obtained from the University of Alberta, Swine Research and Technology Centre (Edmonton, AB, CAN). The piglets were weighed and then pre-anaesthetized with acepromazine (0.5 mg/kg; Atravet™; Ayerst Laboratories, Montreal, PQ); anesthesia was maintained during surgery with 3–4% isoflurane. The piglets (n = 32) had a venous catheter implanted (femoral) and gastric catheters were inserted according to a previous method [25]. A sampling catheter was inserted into the left femoral vein and advanced to the inferior vena cava just caudal to the heart. After surgery, incision sites were treated with a topical antibiotic (Hibitane Veterinary Ointment: Ayerst Laboratories, Montreal, PQ) and an analgesic (0.1 mg/kg Buprenex, Buprenorphrine HCl, Reckitt and Colman Pharmaceutical Inc., Richmond, VA) was given intramuscularly immediately and again 8 h post-surgery. Piglets were then put into cotton jackets, which secured the tether to the piglets. The tether was part of the swivel-tether system (Alice King Chatham Medical Arts, Los Angeles, California), that enabled the pig to move freely while receiving a continuous dietary infusion, ensuring that the catheters did not become tangled or occluded.

### Animal housing

Piglets were housed in individual circular cages, 75 cm in diameter and toys were added to enhance their environment. Piglets weighed 1554 g ± 27 upon arrival and 2539 g ± 6 at treatment initiation. The animal rooms were maintained at an ambient temperature of 25˚C, with supplemental heat supplied by heat lamps. The lighting schedule was 12 h of light commencing at 0600h.

### Diet regimen

Elemental diets were provided as continuous infusions by pressure sensitive infusion pumps. Piglets received 15 g AA kg$^{-1}$·d$^{-1}$ and 1.1 MJ metabolizable energy kg$^{-1}$·d$^{-1}$ with glucose and lipid (Intralipid 20%, Fresenius-Kabi, Stockholm, Sweden) each supplying 50% of nonprotein energy intake. The base AA profile of the complete elemental diet fed during adaptation (d 0 until d 5) has been previously described [4,26]. The AA profile was based on human milk protein (Vaminolact: Fresenius-Kabi, Stockholm, Sweden) except phenylalanine and tyrosine which were provided at their estimated safe levels of intake [27,28] and arginine was provided at 1.2 g·kg$^{-1}$· d$^{-1}$ [29]. Diet infusion rates were adjusted daily after weighing the piglets. Vitamins were supplied as a commercial solution, MVI Pediatric (Rhone-Poulenc Rorer Canada Inc, Montreal, PQ) which was added to the diet immediately prior to feeding. The cofactors involved in the transsulfuration pathway, vitamin B-12, choline, B-6 and folate were provided via the MVI solution at approximately 115% of requirement [6]. Piglets also received a mineral solution including zinc, copper, manganese, chromium, selenium and iodide at >200% of the recommendation for piglets [6]. Iron was supplied as iron dextran solution in the diet solution (16 mg/mL).

The elemental diet was infused as total parenteral nutrition (TPN) immediately following surgery, and increased to full infusion rates (13.5 mL· kg$^{-1}$ ·h$^{-1}$) by the end of day 1 [4]. Piglets were transitioned to enteral feeding in a step-wise procedure which was completed by the end of day 2. Piglets received both diet and isotope enterally. Piglets were then randomly allocated

to one of the eight test levels of Cys (0, 0.05, 0.1, 0.15, 0.2, 0.25, 0.40, 0.50 g kg$^{-1}$ d$^{-1}$) with a constant intake of Met (0.25 g kg$^{-1}$ d$^{-1}$), totaling 4 pigs per dietary cysteine intake. Randomizations were performed using the Microsoft Excel function Rand(). Sample size was decided based on previous studies using isotope dilution techniques in neonatal piglets [4,5]. Cys was provided as L-Cys free base in all test diets. All test diet solutions were made isonitrogenous by altering the concentration of L-alanine. The solutions were sterilized with a 0.22 μm filter (Millipore, Milford, MA). Due to the unstable nature of L-Cys in aqueous solutions, test diets were made immediately prior to infusions. Piglets were maintained on a test diet from 1800h on d 5 until the completion of the second oxidation study and subsequent necropsy on d 8.

## Tracer infusion, $^{14}CO_2$ collection and analytical procedures

With exception of the researcher responsible for the randomization of experimental units, the others were blinded to treatments during the conduct of the experiment. Met and Cys infusion experiments were conducted on d 6 and d 8. On each day, half of the piglets received Met isotope infusion while the other half receive Cys isotope infusion to account for potential confounding effect of the order of isotope. Met oxidation represents Cys synthesis, or the rate of transsulfuration, as the release of 1-carbon of Met can only occur via the transsulfuration pathway. Met and Cys oxidation were determined by a primed (186 kBq (5 μCi/kg)), constant intravenous infusion (186 kBq (5 μCi·kg$^{-1}$·h$^{-1}$)) of a tracer solution containing 92.8 MBq (2.5 mCi)/L of L-[1-$^{14}$C]Met or L-[1-$^{14}$C]Cys (200 MBq (54 mCi/mmol) American Radiolabeled Chemicals, Inc. St. Louis, MO)). The constant infusion was for 6 h, in order to achieve plateau in breath labeling. One hour of background $^{14}CO_2$ collection was taken only on d 8 and background was subtracted from total $^{14}CO_2$ collected during the d 8 infusion; background $^{14}CO_2$ on d 6 was assumed to be zero as $^{14}$C is not produced endogenously. Details of infusion protocol, $^{14}CO_2$ collection and blood collection procedures have been described previously [27]. Following the infusion on d 8, piglets were anesthetized with isoflurane and killed by injection of 1000 mg of sodium pentobarbital into a venous catheter.

## Determination of plasma AA including Hcy

Blood was collected at the end of each infusion experiment as previously described [27] to determine plasma concentrations of AA and Hcy. Plasma AA concentrations were measured using reverse-phase high performance liquid chromatography (HPLC) as phenylisothiocyanate derivatives. Total Hcy and total Cys concentrations were analyzed according to a reverse phase-HPLC method [30] with modifications [31]. Briefly, plasma samples were incubated with tris-carboxyethylphosphine (Pierce Chemicals, Mississauga, ON), to reduce protein-bound and oxidized forms of Cys and Hcy, followed by derivatization with 7-fluorobenzofurazan-4-sulfonic acid ammonium salt (SBD-F; Sigma Chemical Co., Oakville, ON). The fluorescent thiol derivatives were separated on a Waters C-18 column (5 μM, 4.5 x 250 mm; Waters Canada, Mississauga, ON), using isocratic elution (98% 0.1 M acetate, pH 5.5: 2% methanol) by means of a Shimadzu HPLC system (Man-Tech Associates, Guelph, ON) complete with autoinjector and fluorescence detector (excitation λ = 385 nm; emission λ = 515 nm). Concentrations of total Cys and Hcy were determined through the use of an external standard curve, and the inter- and intra-assay coefficients of variation were < 2%.

## Calculations

The rate of $^{14}CO_2$ expiry (dpm kg$^{-1}$ h$^{-1}$) was determined and data were corrected for the retention of label in the bicarbonate pool using a bicarbonate retention factor (BRF) of 0.933 [32].

The resulting equations appear as follows:

$$\text{Corrected V}^{14}\text{CO}_2\left(\text{dpm kg}^{-1}\,\text{h}^{-1}\right) = \frac{\text{V14CO2 (dpm kg} - 1\,\text{h} - 1)}{\text{BRF}} \tag{1}$$

$$\text{Percent of dose oxidized (\%)} = \frac{\text{plateau corrected V14CO2}}{\text{isotope infusion (dpm kg} - 1\,\text{h} - 1)} * 100 \tag{2}$$

### Statistical analyses

Researchers responsible for raw data assessment were blinded to which experimental treatment each piglet received. A fixed effect model with Cys intake serving as the main treatment effect was used. Significant differences in Cys synthesis and oxidation among Cys intakes were determined using an ANOVA. If P values were <0.05 for the F-value of the ANOVA model, significant differences among treatments were determined using the Student Newman Keul's multiple comparison procedure (SAS/STAT, version 8.1, SAS Institute, Cary, NC).

Determination of the dietary intake of Cys required to reduce Cys synthesis (Met oxidation) to obligatory Met oxidation, was performed using a two-way linear crossover model, as described previously [33,34]. Regression analysis variables were dietary concentration of Cys as the independent variable and percentage of Met dose oxidized as the dependent variable. To determine the amount of dietary Cys required to reduce Met oxidation to obligatory Met oxidation levels, the data points were iteratively partitioned between two distinct regression lines. The final partitioning of the data for the two regressions was chosen as the model that produced the highest regression coefficients for the dependent variables. The point at which the two regression lines intersected provides an estimate of the maximum amount of Cys sparing. The 95% confidence intervals, for the estimation of a safe level of intake, were also determined.

The effects of Cys intake on Cys oxidation were analyzed using PROC REG (SAS/STAT, version 8.1, SAS Institute, Cary, NC) and if an effect was defined, we considered the 95% asymptote as the minimal Cys oxidation.

Plasma AA data were analyzed separately for each isotope infusion by ANOVA using the PROC GLIMMIX procedure (SAS version 9.4, SAS Inst., Inc., Cary, NC) with dietary Cys as the fixed effect. Results were considered significant at P < 0.05. Significant differences between treatment means were separated using the Tukey's test. In addition, polynomial contrasts were used to evaluate the response of plasma AA and Hcy concentrations to increasing dietary Cys intake. The IML procedure in SAS was used to generate the coefficients for the unequally spaced linear and quadratic contrasts. The effect of time (day 6 vs day 8) on plasma concentration of amino acids was evaluated using the PROC GLIMMIX procedure (SAS version 9.4, SAS Inst., Inc., Cary, NC) with time as the fixed effect and pig as the random effect. Residual plots and proc univariate were used to check model assumptions for each plasma AA. If assumptions were violated, data were log-transformed and (or) a modification in the covariance structure were performed. Results were considered significant at P < 0.05. Data points were removed from analysis if AA concentrations were outside the biological range.

## Results

### Piglet performance

All piglets were healthy during the course of this study. Body weight did not differ (P>0.05) among dietary Cys levels. In addition, rates of average daily gain for the 5-day test period before test diet initiation did not differ (162 g/d ± 6) (P>0.05).

**Table 1.** $^{14}CO_2$ from L-[1-$^{14}C$]Met and L-[1-$^{14}C$]Cys in piglets receiving total enteral nutrition with graded levels of dietary Cys and 0.25 g kg$^{-1}$ d$^{-1}$ Met[1].

| n | Cys Intake (g·kg$^{-1}$· d$^{-1}$) | | | | | | | | pooled SE | ANOVA P value |
|---|---|---|---|---|---|---|---|---|---|---|
| | 0 | 0.05 | 0.1 | 0.15 | 0.2 | 0.25 | 0.4 | 0.5 | | |
| | 4 | 4 | 4 | 4 | 4 | 4 | 4 | 4 | | |
| percent of Met oxidized[2] | 20.6[a] | 16.0 [ab] | 11.6 [bc] | 7.0 [cd] | 6.1[cd] | 6.0[cd] | 5.1[cd] | 3.4[d] | 1.15 | <0.0001 |
| percent of Cys oxidized | 22.8 | 17.4 | 14.2 | 13.5 | 17.0 | 17.2 | 17.3 | 18.0 | 0.86 | NS[3] |

[1] Values represent the means of 4 pigs per dietary Cys intake. Values represent the percent of dose oxidized at isotopic steady state.

[2] Overall ANOVA, F-test, P<0.05. Values with different superscript letters indicate a significant difference among diet levels (Student Newman Keul's multiple comparisons procedure).

[3] Non-significant (P>0.05).

## Isotope studies

Values for $^{14}CO_2$ recovery for both Met and Cys infusion are summarized in Table 1. No data points were excluded in the analysis. Plateaus in breath $^{14}CO_2$ from 1-$^{14}C$-Met were reached within 4 hours after the initiation of the primed constant infusion in all pigs. Plateaus in breath $^{14}CO_2$ from 1-$^{14}C$-Cys were reached within 1 hour after the initiation of the primed constant infusion in all pigs. Transsulfuration (Met oxidation), expressed as a percentage of the Met dose oxidized during isotopic steady state, was significantly influenced by Cys intake (P<0.0001, Fig 1). As Cys intake increased from 0 to 0.15 g kg$^{-1}$ d$^{-1}$, transsulfuration linearly decreased (slope = -90% dose oxidized/ 100 mg of Cys intake, P = 0.0004). Further increases in Cys intake (from 0.15 to 0.5 g Cys/ kg d$^{-1}$) resulted in a small but significant decrease in transsulfuration (slope = -8% dose oxidized/ 100 mg of Cys intake, P = 0.005); because the slope of this second line was different from zero, increasing Cys intake resulted in a reduction in transsulfuration, but at a slower rate of change than the first regression line. The breakpoint estimate for transsulfuration or Cys synthesis (Fig 1) was 0.15 g kg$^{-1}$ d$^{-1}$ (95% confidence interval: 0.11–0.20 g kg$^{-1}$ d$^{-1}$).

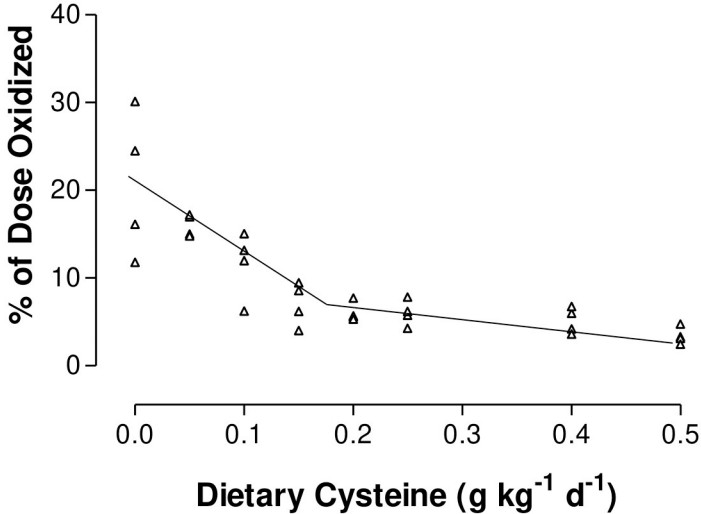

**Fig 1. L-[1-$^{14}C$]Met oxidation as a percentage of dose, representing the change in Cys synthesis, in enterally fed piglets receiving graded levels of Cys and 0.25 g kg$^{-1}$ d$^{-1}$ Met.** The break-point value was 0.15 g kg$^{-1}$ d$^{-1}$ with a confidence interval of 0.11–0.20 g kg$^{-1}$ d$^{-}$.

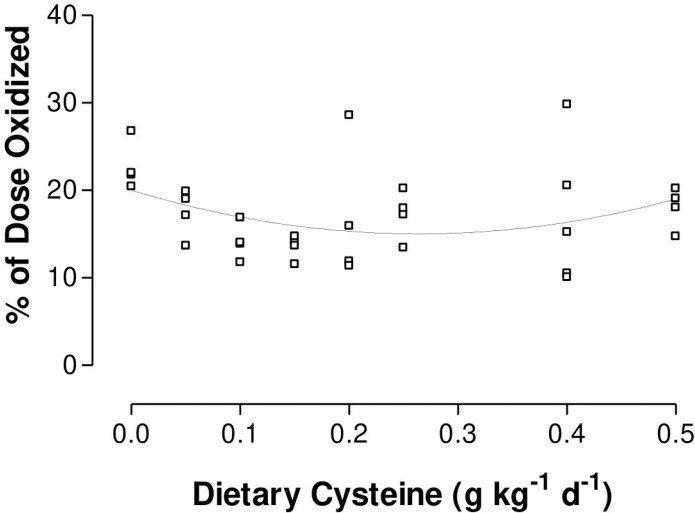

**Fig 2. Oxidation of L-[1-$^{14}$C]Cys as a percentage of dose in enterally fed piglets receiving graded levels of Cys and 0.25 g kg$^{-1}$ d$^{-1}$ Met.** Cys oxidation was associated with Cys intake in a second order polynomial response (y = 20.09–27.79x + 71.70x$^2$; P<0.05, R$^2$ = 0.12) with Cys oxidation minimized at 0.25 g Cys kg$^{-1}$ d$^{-1}$.

There were no differences (P>0.05) in Cys oxidation among dietary Cys intakes during enteral feeding when lsmeans were compared using an ANOVA. When the data were regressed against Cys intake, (Fig 2), there was a significant polynomial relationship (P = 0.0006, R$^2$ = 0.12, MSE = 3.1) between Cys oxidation and Cys intake. Cys oxidation was minimized at 0.25 g kg$^{-1}$ d$^{-1}$ Cys intake.

## Obligatory met oxidation

Met oxidation at Cys intakes above the breakpoint were assumed to represent an estimate of obligatory Met oxidation. The mean obligatory Met oxidation during enteral feeding (n = 16) was 5.16 ± 1.58 (SD).

## Plasma concentrations of AA

Plasma concentrations of AA were assessed after infusion of L-[1-$^{14}$C]Met (Table 2) and L-[1-$^{14}$C]Cys (Table 3).

## Plasma concentrations of AA on L-[1-$^{14}$C]Met infusion study day

Plasma concentrations of Met increased linearly (P < 0.05) with increasing intake of Cys, with greater (P < 0.05) concentrations observed at the highest Cys intake (0.5 g kg$^{-1}$ d$^{-1}$) compared to 0 g kg$^{-1}$ d$^{-1}$ Cys. A cubic response was observed for plasma concentrations of total Cys, with highest concentrations at the highest dietary intakes of Cys (0.4 and 0.5 g kg$^{-1}$ d$^{-1}$) compared to lower intake levels. A cubic response was also observed for total Hcy concentrations with higher (P < 0.05) concentrations at 0.4 g kg$^{-1}$ d$^{-1}$Cys compared to 0, 0.05, 0.1 and 0.5 g kg$^{-1}$ d$^{-1}$ Cys. Taurine concentrations changed in a quadratic fashion (P < 0.05) with higher concentrations at the 0.5 g kg$^{-1}$ d$^{-1}$ Cys compared to 0.1, 0.15 and 0.2 g kg$^{-1}$ d$^{-1}$ Cys.

A linear relationship (P < 0.05) was observed between dietary Cys intake and plasma concentrations of threonine, alanine, citrulline, and hydroxyproline. Plasma threonine concentrations were higher (P < 0.05) at the greatest Cys intake (0.5 g kg$^{-1}$ d$^{-1}$) while hydroxyproline concentrations were higher (P < 0.05) at 0.4 g kg$^{-1}$ d$^{-1}$ Cys compared to 0, 0.05 0.1, and 0.2 g

**Table 2. Plasma amino acid concentrations in piglets receiving total enteral nutrition after L-[1-$^{14}$C]Met infusion.**

| Item[1] | | Cysteine Intake (g·kg-1· d-1) | | | | | | | | SEM[2] | P-value | | | |
|---|---|---|---|---|---|---|---|---|---|---|---|---|---|---|
| | | 0 | 0.05 | 0.1 | 0.15 | 0.2 | 0.25 | 0.4 | 0.5 | | ANOVA | Linear | Quadratic | Cubic |
| IDAA, umol/L | Arg | 187 | 157 | 135 | 183 | 140 | 143 | 158 | 124 | 29 | 0.617 | 0.189 | 0.825 | 0.319 |
| | His | 46 | 56 | 29 | 29 | 24 | 22 | 35 | 51 | 13 | 0.469 | 0.955 | **0.027** | 0.863 |
| | Ile | 144 | 119 | 101 | 118 | 93 | 86 | 105 | 98 | 18 | 0.301 | 0.068 | 0.088 | 0.434 |
| | Leu* | 288 | 255 | 214 | 266 | 213 | 215 | 289 | 208 | 51 | 0.697 | 0.527 | 0.613 | 0.178 |
| | Lys | 524 | 653 | 406 | 503 | 356 | 335 | 574 | 474 | 79 | **0.047** | 0.540 | 0.037 | 0.407 |
| | Met | 33$^b$ | 30$^b$ | 42$^{ab}$ | 45$^{ab}$ | 40$^{ab}$ | 36$^{ab}$ | 46$^{ab}$ | 73$^a$ | 9.2 | **0.027** | **0.001** | 0.188 | 0.188 |
| | Phe | 38 | 37 | 44 | 39 | 23 | 21 | 26 | 56 | 13 | 0.385 | 0.742 | 0.055 | 0.095 |
| | Thr* | 469$^b$ | 437$^b$ | 465$^b$ | 639$^{ab}$ | 482$^b$ | 543$^b$ | 747$^{ab}$ | 1515$^a$ | 285 | **0.002** | **< .0001** | 0.065 | 0.243 |
| | Trp | 58 | 55 | 31 | 36 | 33 | 25 | 52 | 65 | 13 | 0.174 | 0.419 | **0.006** | 0.616 |
| | Val*# | 446 | 83 | 175 | 206 | 160 | 154 | 190 | 183 | 257 | 0.533 | 0.728 | 0.310 | 0.190 |
| DAA, umol/L | Ala | 442 | 405 | 361 | 445 | 379 | 382 | 565 | 581 | 66 | 0.063 | **0.006** | 0.078 | 0.402 |
| | Asp* | 7.9 | 11 | 7.4 | 6.9 | 7.2 | 5.5 | 11.7 | 15.3 | 3.7 | 0.134 | 0.057 | **0.027** | 0.779 |
| | Cit | 139 | 115 | 86 | 97 | 82 | 125 | 55 | 77 | 31 | 0.172 | **0.059** | 0.585 | 0.719 |
| | Glu | 92 | 86 | 102 | 82 | 109 | 64 | 79 | 136 | 24 | 0.586 | 0.389 | 0.238 | 0.174 |
| | Gly | 908$^{abc}$ | 767$^{bc}$ | 548$^c$ | 607$^c$ | 549$^c$ | 602$^c$ | 1237$^{ab}$ | 1368$^a$ | 148 | **0.0002** | 0.0001 | **0.0003** | 0.082 |
| | Gln | 369$^a$ | 272$^{ab}$ | 225$^{abc}$ | 238$^{abc}$ | 180$^{bc}$ | 239$^{abc}$ | 117$^c$ | 299$^{ab}$ | 34 | **0.001** | 0.028 | **0.001** | 0.264 |
| | Ohp | 70$^b$ | 61$^b$ | 64$^b$ | 92$^{ab}$ | 70$^b$ | 69$^b$ | 117$^a$ | 87$^{ab}$ | 11 | **0.002** | **0.002** | 0.833 | 0.048 |
| | Orn* | 229 | 246 | 170 | 210 | 174 | 183 | 209 | 158 | 43 | 0.671 | 0.202 | 0.680 | 0.312 |
| | Pro | 570$^a$ | 322$^{ab}$ | 346$^{ab}$ | 441$^{ab}$ | 277$^b$ | 362$^{ab}$ | 456$^{ab}$ | 550$^a$ | 65 | **0.008** | 0.178 | **0.002** | 0.217 |
| | Ser | 258$^{ab}$ | 260$^{ab}$ | 171$^b$ | 180$^b$ | 174$^b$ | 162$^b$ | 254$^{ab}$ | 333$^a$ | 27 | **0.000** | 0.006 | **< .0001** | 0.715 |
| | Tau | 142$^{ab}$ | 138$^{ab}$ | 117$^b$ | 111$^b$ | 95$^b$ | 108$^b$ | 169$^{ab}$ | 219$^a$ | 20 | **0.004** | 0.002 | **0.001** | 0.905 |
| | Total Cys | 32$^{cd}$ | 27$^d$ | 37$^{cd}$ | 47$^{cd}$ | 66$^{bc}$ | 80$^b$ | 141$^a$ | 123$^a$ | 8.0 | **< .0001** | **< .0001** | 0.797 | **0.0001** |
| | Total Hcy | 4.9$^b$ | 6.3$^b$ | 7.2$^b$ | 9.1$^{ab}$ | 8.5$^{ab}$ | 9.6$^{ab}$ | 14.0$^a$ | 6.0$^b$ | 1.5 | **0.003** | 0.018 | 0.003 | **0.015** |
| | Tyr* | 86$^a$ | 66$^{ab}$ | 44$^{abc}$ | 26$^{abc}$ | 17$^c$ | 20$^{bc}$ | 29$^{abc}$ | 60$^{ab}$ | 22 | **0.001** | 0.104 | **< .0001** | 0.767 |

$^{a,b,c}$ Values in the same row followed by different superscripts differ significantly.

[1]IDAA = indispensable amino acid; DAA = dispensable amino acid.

[2]Standard error of the mean.

*Log-transformed.

#Covariance structure modified.

^No significant differences were observed between treatments when pairwise comparisons were evaluated using the Tukey adjustment for control of type I error. I.

kg$^{-1}$ d$^{-1}$ Cys. A quadratic response (P < 0.05) was observed for plasma concentrations of histidine, tryptophan, aspartate, glycine, glutamate, proline, serine, and tyrosine. A U-shaped response was observed for the concentrations of the aforementioned AAs, with highest concentrations at the lowest and highest intake levels of Cys.

**Plasma concentrations of AA on L-[1-$^{14}$C]Cys infusion study day.** Met concentrations also increased linearly (P < 0.05) with increasing Cys intake after [1-$^{14}$C]Cys infusion and similar to that observed with the Met infusion. A cubic response (P < 0.05) was observed for plasma concentrations of total Cys with highest concentrations at the highest dietary intakes of Cys (0.4 and 0.5 g kg$^{-1}$ d$^{-1}$) and similar to that observed with the Met infusion. Taurine and total Hcy concentrations changed in a quadratic fashion (P < 0.05) with increasing intakes of Cys and similar to that observed with the Met infusion.

Phenylalanine increased linearly (P < 0.05) while a quadratic response was observed for valine, glycine, proline, serine, and tyrosine with increasing levels of Cys intake (P < 0.05). A cubic response (P < 0.05) was observed for plasma concentrations of threonine, glutamine

**Table 3. Plasma amino acid concentrations (lsmeans) in piglets receiving total enteral nutrition after L-[1-[14]C]Cys infusion.**

| Item[1] | | Cysteine Intake (g·kg-1· d-1) | | | | | | | | SEM[2] | P-value | | | |
|---|---|---|---|---|---|---|---|---|---|---|---|---|---|---|
| | | 0 | 0.05 | 0.1 | 0.15 | 0.2 | 0.25 | 0.4 | 0.5 | | ANOVA | Linear | Quadratic | Cubic |
| IDAA, umol/L | Arg | 177 | 176 | 181 | 185 | 168 | 135 | 183 | 154 | 25 | 0.840 | 0.476 | 0.808 | 0.974 |
| | His | 72 | 69 | 68 | 78 | 45 | 43 | 53 | 75 | 17 | 0.582 | 0.634 | 0.143 | 0.285 |
| | Ile | 146 | 126 | 130 | 125 | 119 | 108 | 119 | 110 | 16 | 0.773 | 0.131 | 0.400 | 0.663 |
| | Leu | 244 | 263 | 272 | 264 | 246 | 235 | 261 | 227 | 33 | 0.974 | 0.543 | 0.750 | 0.910 |
| | Lys | 818 | 831 | 804 | 718 | 561 | 653 | 862 | 934 | 118 | 0.405 | 0.431 | 0.040 | 0.985 |
| | Met | 34 | 40 | 33 | 37 | 27 | 41 | 46 | 65 | 7.8 | 0.074 | **0.005** | 0.064 | 0.567 |
| | Phe* | 24[b] | 49[ab] | 55[ab] | 56[ab] | 41[b] | 31[b] | 41[b] | 88[a] | 26 | **0.008** | **0.009** | 0.141 | 0.001 |
| | Thr | 431[b] | 590[b] | 605[b] | 650[b] | 542[b] | 759[b] | 915[b] | 2198[a] | 111 | **< .0001** | **< .0001** | **< .0001** | **0.001** |
| | Val* | 965 | 165 | 246 | 230 | 193 | 185 | 233 | 408 | 340 | 0.207 | 0.623 | **0.007** | 0.152 |
| DAA, umol/L | Ala | 471 | 439 | 448 | 509 | 439 | 494 | 649 | 904 | 72 | **0.012** | 0.058 | 0.859 | 0.859 |
| | Asp | 14 | 16 | 13 | 15 | 19 | 14 | 14 | 18 | 3.4 | 0.893 | 0.575 | 0.822 | 0.471 |
| | Cit | 159 | 122 | 115 | 106 | 86 | 96 | 66 | 39 | 44 | 0.370 | 0.019 | 0.736 | 0.571 |
| | Glu | 94 | 131 | 114 | 109 | 151 | 140 | 102 | 166 | 27 | 0.496 | 0.199 | 0.905 | 0.178 |
| | Gln | 387[a] | 330[a] | 315[ab] | 276[ab] | 236[ab] | 214[ab] | 122[b] | 365[a] | 45 | **0.004** | 0.049 | 0.001 | **0.015** |
| | Gly | 1239 | 785 | 625 | 777 | 860 | 887 | 1166 | 1541 | 169 | **0.013** | 0.007 | **0.004** | 0.215 |
| | Ohp | 69[dc] | 64[d] | 73[dc] | 88b[dc] | 80[dc] | 97[abc] | 121[a] | 114[ab] | 6.8 | **< .0001** | **< .0001** | 0.520 | **0.057** |
| | Orn | 284 | 316 | 351 | 298 | 231 | 263 | 264 | 329 | 34 | 0.272 | 0.781 | 0.143 | 0.074 |
| | Pro | 567[ab] | 403[ab] | 460[ab] | 463[ab] | 281[b] | 400[aba] | 533[b] | 636[a] | 65 | **0.021** | 0.089 | **0.002** | 0.606 |
| | Ser | 256 | 299 | 228 | 217 | 227 | 221 | 263 | 386 | 37 | 0.054 | 0.037 | **0.006** | 0.330 |
| | Tau | 162 | 161 | 155 | 154 | 118 | 134 | 162 | 252 | 26 | 0.052 | 0.028 | **0.006** | 0.196 |
| | Total Cys | 44[cb] | 34[b] | 43[cb] | 58[cb] | 67[cb] | 82[b] | 126[a] | 134[a] | 9.1 | **< .0001** | **< .0001** | 0.356 | **0.038** |
| | Total Hcy | 7.8 | 6.8 | 9.3 | 10 | 7.1 | 8.2 | 9.6 | 4.8 | 1.33 | 0.070 | 0.246 | **0.060** | 0.292 |
| | Tyr | 46 | 65 | 44 | 33 | 29 | 16 | 34 | 59 | 12 | 0.069 | 0.721 | **0.006** | 0.162 |

[a,b,c] Values in the same row followed by different superscripts differ significantly.

[1] IDAA = indispensable amino acid; DAA = dispensable amino acid.

[2] Standard error of the mean.

and hydroxyproline. Similarly to what was reported for plasma AA concentrations after [1-[14]C]Met infusion, most AAs had a U-shaped response with increasing level of Cys intake—where highest concentrations were observed at the extreme ends (lowest and highest intake level).

**Plasma concentrations of AAs over time.** Plasma concentrations of arginine, leucine, lysine, valine, glutamine, glutamate, and taurine were lower ($P < 0.05$) on day 8 compared to day 6 (Table 4). No significant effect ($P > 0.05$) of time was observed for plasma concentrations of the other amino acids.

## Discussion

To our knowledge, this is the first *in vivo* examination of the effects of increasing dietary Cys, with a constant Met supply, on both Cys synthesis (i.e., Met oxidation) and Cys oxidation in young pigs. This study clearly demonstrated that as Cys intake increased, Cys synthesis (i.e., transsulfuration, as measured by Met oxidation) decreased in a linear fashion until the dietary requirement for Cys was met or exceeded (Fig 1). Cys synthesis decreased until the sum of dietary Met (0.25 g kg[-1] d[-1]) and Cys (0.15 g kg[-1] d[-1]) equaled 0.40 g kg[-1] d[-1] TSAA intake. This decrease in Cys synthesis with increasing Cys intake between 0 and 0.15 g kg[-1] d[-1] represents

**Table 4. The effect of sampling time (day 6 vs. day 8) on plasma amino acid concentrations (lsmeans + SEM[1]).**

| Item[2] | | Day | | P-value |
|---|---|---|---|---|
| | | 6 | 8 | |
| IDAA, umol/L | Arg | 178 + 8.52 | 148 + 8.38 | **0.009** |
| | His | 48 + 5.60 | 50 + 5.59 | 0.808 |
| | Ile | 123 + 5.66 | 112 + 5.57 | 0.106 |
| | Leu | 286 + 14 | 233 + 14 | **0.001** |
| | Lys | 669 + 44 | 518 + 43 | **0.017** |
| | Met | 43 + 3.87 | 47 + 3.81 | 0.410 |
| | Phe | 39 + 4.28 | 41 + 4.2 | 0.628 |
| | Thr | 754 + 89 | 758 + 88 | 0.947 |
| | Val | 370 + 65 | 237 + 64 | **0.036** |
| DAA, umol/L | Ala | 510 + 29 | 472 + 29 | 0.281 |
| | Asp | 14 + 1.09 | 11 + 1.07 | 0.123 |
| | Cit | 106 + 10 | 91 + 10 | 0.125 |
| | Glu | 131 + 8.67 | 92 + 8.54 | **0.001** |
| | Gln | 281 + 15 | 245 + 18 | **0.024** |
| | Gly | 910 + 73 | 908 + 72 | 0.976 |
| | Ohp | 88 + 4.38 | 80 + 4.33 | 0.081 |
| | Orn | 250 + 28 | 290 + 28 | 0.320 |
| | Pro | 454 + 27 | 431 + 27 | 0.450 |
| | Ser | 256 + 14 | 233 + 14 | 0.120 |
| | Tau | 161 + 10.08 | 140 + 9.96 | **0.039** |
| | Total Cys | 77 + 8.19 | 71 + 8.22 | 0.176 |
| | Total Hcy | 8.32 + 0.57 | 8.0 + 0.58 | 0.544 |
| | Tyr | 48 + 9.96 | 53 + 9.96 | 0.238 |

[1]Standard error of the mean.

[2]IDAA = indispensable amino acid; DAA = dispensable amino acid.

Cys sparing the Met requirement for protein synthesis. This sparing effect is a result of the redistribution of Hcy between the remethylation and transsulfuration pathways. A notable decrease in the latter is observed through the inhibition of the liver enzymes and a reduction in SAM, which is an allosteric inhibitor of methylene-tetrahydrofolate reductase [35] and an activator of CBS, the first enzyme in the transsulfuration pathway [36]. Controversially, previous studies using human fetal tissues reported a lack of activity of cystathionine γ-lyase (CGL), which is the final enzyme required for cysteine synthesis in the transsulfuration pathway [37,38]. A lack of functional CGL would limit the sparing effect of Cys in this population. However, later studies demonstrated the ability of preterm neonates to synthesize Cys through the transsulfuration pathway [39,40]. This is in agreement with a previous author [41] who showed that the hepatic activity of CBS and CGL gradually increase with age in piglets, plateauing post-weaning at 18 d of age. The sparing effect only occurs when Met is provided at levels above its minimum requirement and at less than the TSAA requirement [42]. As we held the Met intake within this range (50% of the recommended TSAA requirement), we were able to capture the sparing effect of Cys in piglets in the current study.

The present estimate of 0.40 g kg$^{-1}$ d$^{-1}$ for the TSAA requirement, using minimum transsulfuration, is similar to our previous estimate of the mean TSAA requirement (0.42 g kg$^{-1}$ d$^{-1}$) determined by indicator AA oxidation in neonatal piglets receiving an enteral diet providing

Met only [4]. If this present estimate is corrected on a molar basis, it represents 0.44 g kg$^{-1}$ d$^{-1}$ Met equivalents. These data also agree with previous data demonstrating that Met oxidation (i.e., Cys synthesis) was reduced when Cys partially replaced dietary Met in adult humans receiving an oral diet [23]. Similarly, the intake of excess Cys reduced the Met requirement in school-age children [43].

The Met requirement is not static and may alter depending on environmental and physiological conditions, and dietary factors. For example, methyl donors have the ability to contribute to the Met requirement in neonates [44]. Attention should also be paid to the supply of dietary co-factors involved in the Met cycle, such vitamin B6, as their deficiency can greatly impact the metabolism of SAA [45]. Furthermore, balanced Met:Cys ratio is imperative to support the endogenous production of Cys and glutathione [46], which have an effect on inflammation, and thus, may affect the Met requirement. Regarding the effects of physiological and pathological conditions on the requirement of Met and TSAA, the TSAA requirement increases when piglets face a bacterial challenge [47–49], likely because of a higher metabolic need for glutathione due to the increased oxidative stress [50]. As such, the metabolic fluxes involved in the Met cycle will be altered. The Met:Cys ratio in the diet is also important. For example, in rats facing a bacterial challenge, a balanced dietary Met:Cys ratio (50Met:50Cys) increased concentrations of glutathione compared to rats fed an imbalanced Met:Cys ratio (100Met:100Cys) [51]. Thus, the investigation of Met and Cys kinetics under different dietary regimens and disease states is highly warranted.

Increases in Cys intake from 0.15 to 0.50 g kg$^{-1}$ d$^{-1}$, resulted in a small, but significant, reduction in transsulfuration; there are several possible explanations for this response. This reduction may be due to population variation among piglets in the TSAA requirement, resulting in detection of a small sparing effect beyond the population mean requirement for the TSAA. However, the 95% confidence interval (0.11–0.20 g Cys kg$^{-1}$ d$^{-1}$) for the mean requirement (0.15 g Cys kg$^{-1}$ d$^{-1}$) suggests that either the population distribution of the Met requirement is skewed dramatically to the right (i.e. is greater than 3 times the mean requirement), which is unlikely, or that some other explanation is required. Alternatively, the second regression line may represent the effect of excess Cys on the regulation of the transsulfuration pathway. Other researchers have shown that rats fed a Cys supplemented diet had lower *in vitro* hepatic CBS activity compared to rats fed a diet containing Met alone [15,52]. The number of Cys intakes levels (n = 4) and corresponding data points (n = 16) above the TSAA requirement may have made the present experiment sensitive enough to detect this small effect on the second regression line (Fig 2). Many other experiments have examined dietary Cys intakes at or above the requirement, which would fall on the second, less pronounced, regression line [7–12]. Because the response to dietary Cys intakes above the TSAA requirement is very small, experiments using fewer diets and subjects may not detect differences among dietary Cys intakes.

These data also demonstrate that when the TSAA intake exceeded the requirement, using Cys supplementation and Met at 50% of the TSAA requirement, ~5–8% of dietary Met intake was still oxidized. Although the present study cannot identify whether the obligatory oxidation of Met was via transsulfuration or transamination [53], it is likely via transsulfuration because the presence and significance of the transamination pathway has not been demonstrated in pigs or humans. Because the transsulfuration pathway is the primary method of disposing the sulfur moiety of Met [54], the basal rate of transsulfuration is likely maintained simply for Met catabolism, rather than for Cys synthesis. However, as the transsulfuration rate is directly impacted by the Met and Cys intake, the true basal rate of transsulfuration can only be assessed when a TSAA-free diet is provided. Our results suggest that the basal rate of transsulfuration remains at a minimum of 5–8% of Met intake when Met is provided at an adequate intake of 50% of the TSAA requirement in neonatal piglets.

There was a significant second order polynomial response (P = 0.0006; Fig 2), where Cys oxidation was initially high, was reduced to a value slightly greater than the TSAA requirement (~ 0.25 g kg$^{-1}$ d$^{-1}$) and then increased again with higher Cys intakes. The higher rate of Cys oxidation at low Cys intakes was possibly due to the upregulation of CDO. In rats fed a diet containing adequate Met, but no Cys, a higher hepatic CDO activity was observed compared to rats fed a diet containing supplemental Cys [55]. Thus, it appears that when Cys and the TSAA limit protein synthesis, Cys oxidation is increased by up-regulation of hepatic CDO activity. However, as Cys intake increased and excess AAs decreased, Cys oxidation decreased until the requirement for the TSAA was met. Supplemental dietary Cys, when compared to a basal diet, results in higher activity of hepatic CDO in rats [55,56]. Furthermore, CDO activity increased in a dose-response manner in hepatocytes from rats cultured in either Met or Cys supplemented media [57,58]. However, when Cys oxidation was compared among the diets by an ANOVA in the present study, we found no differences (Table 1). The second order polynomial response detected within the present study may only have been detectable because of the high numbers of both piglets and dietary Cys intakes studied.

The majority of AAs that showed differences in plasma concentrations at different Cys intake levels had a similar response, with higher plasma concentrations at deficient levels of Cys intake and lower concentrations at Cys intake close to the TSAA requirement. First, during acute AA deficiency, muscle protein synthesis is reduced, and AAs are released into the systemic circulation. Furthermore, the contribution to the pool of free AA from protein catabolism increases relative to their uptake for protein synthesis, which leads to an increase in their plasma concentrations. On the other hand, when the requirement of the limiting AA has been fulfilled, the other AAs are shunted to protein synthesis leading to a decrease in their concentrations in plasma. Controversially, plasma concentrations of hydroxyproline, which is released during collagen degradation and used as an indicator of protein breakdown [59], were greater at the two highest intakes of Cys (P < 0.05). This may indicate a greater protein turnover as all AAs were over the requirement.

Plasma concentrations of Met after L[1-$^{14}$C]Cys infusion were higher at 0.2 g kg$^{-1}$ d$^{-1}$ Cys intake level compared to 0 g kg$^{-1}$ d$^{-1}$ Cys intake which is in agreement with the Met oxidation results. Transsulfuration decreased with increasing Cys intake between 0 and 0.15 g kg$^{-1}$ d$^{-1}$ which explains the increase in plasma Met concentrations. At 0.4 g kg$^{-1}$ d$^{-1}$ Cys intake, Met concentrations decreased again which indicates a higher incorporation of Met into protein after the requirement of TSAA was met. Although a significant second order polynomial response was observed for Cys oxidation, with lower oxidation at 0.25 g kg$^{-1}$ d$^{-1}$ Cys intake, plasma concentrations of total Cys increased with graded levels of dietary Cys; wherein, plateau values were achieved at highest intake levels (0.4 and 0.5 g kg$^{-1}$ d$^{-1}$ Cys). Perhaps, the higher Cys oxidation observed in the lowest and highest intake levels of Cys in the second order polynomial response were too subtle to impact plasma concentrations of total Cys. However, taurine concentrations changed in a quadratic manner, with greater concentrations observed at the lowest and highest levels of Cys intake, which indicates a higher rate of Cys oxidation and this is in agreement with the Cys oxidation results.

Plasma concentrations of Hcy were different after L-[1-$^{14}$C]Met infusion probably due to the higher variation observed in samples after L-[1-$^{14}$C]Cys infusion (1.4 vs. 1.7, SEM). Although the sample size per dietary cysteine intake (n = 4) is commonly used in isotope kinetic studies due to their high sensitivity, a higher sample size is warranted in future studies aiming to investigate the effect of dietary interventions on plasma AA concentrations. Due to the invasive procedure of the isotope methodology used herein, the number of animals were limited to the minimum as plasma AA concentrations were secondary outcomes in this trial. However, we still observed a similar response pattern was observed among infusions. After L-

[1-$^{14}$C]Met infusion, Hcy concentrations changed in a cubic manner with lower concentrations observed at 0.5 compared to 0.4 g kg$^{-1}$ d$^{-1}$ Cys intake. This suggests that the proportion of Hcy that was remethylated increased compared to the proportion that was transsulfurated, further supporting the sparing effect of Cys on Met. In humans, plasma concentrations of Hcy are associated with cardiovascular risk [60] and are suggested to be directly associated with dietary Met intake. Furthermore, hyperhomocysteinemia provoked by dietary Met intake may lead to oxidative stress in the rat brain, and consequently, increase anxiety-related behavior in this species [61]. While the scope of this research was to prove the fundamental concept of the Cys-sparing effect on Met, future research should investigate the ideal Met:Cys ratio to minimize production of Hcy in neonates while still meeting TSAA and minimum Met requirements, which are essential to proper growth and overall health.

The significant decrease of plasma concentrations of some essential and non-essential AA between day 6 and day 8, may indicate an increased catabolism in the small intestine and/or greater utilization by extraintestinal tissues. A decrease in alanine, glutamine, isoleucine, leucine, threonine and valine in 7 to 21-d-old pigs compared with 1 to 3-d-old pigs was previously reported [62]. Even though the age difference was only 2 d in the current study, it may reflect similar metabolic adaptations. This decrease in plasma AA concentrations may be also a result of hemodilution due to continuous dietary infusion over time. It's noteworthy that plasma AA concentrations offer only a snapshot of what is occurring in the whole body, and thus, do not reflect the kinetic or dynamic aspects of AA metabolism. In order to investigate the effects of different levels of Cys on sulfur amino acid metabolism, we used radioactive isotope-labeled tracers (L-[1-$^{14}$C]Met and L-[1-$^{14}$C]Cys). Thus, the latter provides a better representation of sulfur amino acid kinetics and plasma concentrations should be interpreted with caution.

## Conclusion

In the present study, increasing Cys intake in enterally fed piglets was associated with Cys oxidation in a second order polynomial relationship. We further quantified the rate of Cys synthesis (i.e., transsulfuration) over a range of Cys intakes. Increasing dietary Cys reduced Cys synthesis via transsulfuration, as measured by Met oxidation, until the requirement for the TSAA was met. These data indicate that Cys spares the Met requirement for protein synthesis by reducing Met conversion to Cys. This repartitioning of Met from Cys synthesis could also make Met more available for non-protein pathways such as for the synthesis of transmethylated products. In subjects where health and longevity are the outcomes of interest, one may decrease the dietary supply of Met down to the min Met requirement and provide dietary Cys to meet the remaining of the TSAA requirement to ameliorate the possible detrimental effects of hyperhomocysteinemia associated with oversupply of dietary Met.

## Supporting information

**S1 File. Shoveller_rawdata.**
(XLSX)

## Author Contributions

**Conceptualization:** Anna K. Shoveller, James D. House, Robert F. Bertolo, Paul B. Pencharz, Ronald O. Ball.

**Data curation:** Anna K. Shoveller.

**Formal analysis:** Anna K. Shoveller, Julia G. Pezzali.

**Funding acquisition:** James D. House, Paul B. Pencharz, Ronald O. Ball.

**Investigation:** Anna K. Shoveller, Robert F. Bertolo, Paul B. Pencharz.

**Methodology:** Anna K. Shoveller.

**Project administration:** Robert F. Bertolo.

**Resources:** James D. House.

**Supervision:** James D. House, Robert F. Bertolo, Ronald O. Ball.

**Validation:** Anna K. Shoveller, Julia G. Pezzali.

**Visualization:** Anna K. Shoveller, Julia G. Pezzali.

**Writing – original draft:** Anna K. Shoveller, Julia G. Pezzali.

**Writing – review & editing:** Anna K. Shoveller, James D. House, Robert F. Bertolo, Paul B. Pencharz, Ronald O. Ball.

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
