## [Decision Letter · Decision Letter 0]

12 Jul 2022

PONE-D-22-13606Methionine and cysteine oxidation are regulated in a dose dependent manner by dietary Cys intake in neonatal piglets receiving enteral nutritionPLOS ONE

Dear Dr. Shoveller,

Thank you for submitting your manuscript to PLOS ONE. After careful consideration, we feel that it has merit but does not fully meet PLOS ONE’s publication criteria as it currently stands. Therefore, we invite you to submit a revised version of the manuscript that addresses the points raised during the review process.

- please do follow directions provided by reviewers to improve your manuscript- discuss the relationship of hypermethionine diet on brain oxidative stress and consecutive behavioral changes - clearly emphasize the limitations of the study in the discussion section 

We look forward to receiving your revised manuscript.

Kind regards,

Dragan Hrncic

Academic Editor

PLOS ONE

Journal Requirements:

2. Please include the approval number provided by your ethics committee in your ethics statement in your Methods.

3. As part of your revision, please complete and submit a copy of the Full ARRIVE 2.0 Guidelines checklist, a document that aims to improve experimental reporting and reproducibility of animal studies for purposes of post-publication data analysis and reproducibility: https://arriveguidelines.org/sites/arrive/files/Author%20Checklist%20-%20Full.pdf (PDF). Please include your completed checklist as a Supporting Information file. Note that if your paper is accepted for publication, this checklist will be published as part of your article.

   "This work was supported by grants from the Alberta Pork, Alberta Agricultural Research Institute, (Canadian Institutes of Health Research Fund # 12928) and the Natural Sciences and Engineering Research Council of Canada. (JDH)

Alberta Pork: https://www.albertapork.com/

Canadian Institutes of Health Research:https://cihr-irsc.gc.ca/e/193.html

Natural Sciences and Engineering Research Council of Canada:

  " ext-link-type="uri" xlink:type="simple">https://www.nserc-crsng.gc.ca/Index_eng.asp"  

Reviewers' comments:

Reviewer's Responses to Questions

**Comments to the Author**

1. Is the manuscript technically sound, and do the data support the conclusions?

Reviewer #1: Yes

Reviewer #2: Yes

2. Has the statistical analysis been performed appropriately and rigorously? 

Reviewer #1: Yes

Reviewer #2: No

3. Have the authors made all data underlying the findings in their manuscript fully available?

Reviewer #1: Yes

Reviewer #2: Yes

4. Is the manuscript presented in an intelligible fashion and written in standard English?

Reviewer #1: Yes

Reviewer #2: Yes

5. Review Comments to the Author

Reviewer #1: This article represent original scientific research investigating methionine and cysteine oxidation regulation by dietary cysteine in neonatal piglets receiving enteral nutrition. The article contains all the necessary parts of the scientific work. In the introduction sulfur containing amino acid metabolism is adequatly presented, although the abbreviation for homocysteine should be standardized from hCys to Hcy. It is recomended for authors to provide the clinical implications of this study, in the conclusion section. The literature should be revised and some of the existing references (at least 10) should be replaced with recently published ones (most of them are older than 10 years).

Reviewer #2: General comments.

Methionine and cysteine oxidation was measured at various levels of dietary doses Cys in neonatal pigs to quantify “Cys-sparing effects” on Met oxidation. Inferences based on enzymatic assays and animal growth responses have estimated Cys-sparing effects in animals, however this is the first direct report to quantify Cys sparing effects based on changes in Met oxidation in neonatal pigs (or other animals to my knowledge). The methods and experimental design and techniques are well described and sufficient to answer the questions proposed. As noted in the following specific comments, alternate statistical analysis may provide more quantitative values, although the overall inferences are not likely to be altered. Using 1-14C Met and 1-14C Cys, strong data are reported to support that the Cys-sparing effect occurs by a dose-response inhibition of Met oxidation through the transsulfuration pathway.

The manuscript and research effort is quite refreshing and contributes valuable information on sulfur amino acid metabolism.

Specific Comments.

L 27. Should be, “Methionine (Met) is an…..”

L 75 (and throughout the text). Sentences should be restated to avoid use of phrases “It has been demonstrated”; or “it was found that” (L 79, L 81, L 414, L 416, L 422

L 93 Would the lack of a reduction in Met oxidation in the presence of excess Cys infer a coupling of Cys synthesis to Met oxidation – ie., does a preferential conversion of homocysteine to Cys occur even if Cys is in excess.

L 120. Correct "Piglets weighed" not "weighted"

L 195. Responses fitted to a non-linear model would provide greater statistical robustness that the multiple comparison approach used, although the over inferences may not change.

L 202. Were the iterative partitions subjective or quantitative using a linear-plateau model?

L 297. Responses seem to be a “lack of fit”. A more rigorous statistical analysis is needed for these inferences. Were variances normally distributed across the range of Cys inputs? This concern also affects inferences on Tau and Thr concentrations. Are the quadratic responses being “detected” because of a greater variance on results ate the extremes of Cys intakes, ie., (L 310 to 311). Also L 446 to 449).

L 387 to 389. Perhaps another explanation relates to the statistical models resulting in a “lack of fit”, transformation of the data to adjust for non-normal distributions of variances may eliminate the responses that are somewhat difficult to explain.

6. PLOS authors have the option to publish the peer review history of their article (what does this mean?). If published, this will include your full peer review and any attached files.

Reviewer #1: No

Reviewer #2: No

---

## [Author Response · Author response to Decision Letter 0]

15 Aug 2022

Reviewer #1: This article represent original scientific research investigating methionine and cysteine oxidation regulation by dietary cysteine in neonatal piglets receiving enteral nutrition. The article contains all the necessary parts of the scientific work. In the introduction sulfur containing amino acid metabolism is adequatly presented, although the abbreviation for homocysteine should be standardized from hCys to Hcy. It is recomended for authors to provide the clinical implications of this study, in the conclusion section. The literature should be revised and some of the existing references (at least 10) should be replaced with recently published ones (most of them are older than 10 years).

Dear Reviewer, 

We greatly appreciate your suggestions. As suggested, hCys was replaced by Hcys. We understand that many references are old; however, they are the original research that underpinned the present investigation and as such, are appropriate. However, many of those are necessary as we refer to fundamental work and we must cite the original reference. Many of the work cited refers to the first discoveries pertaining to sulfur amino acid metabolism, and thus, they are old by nature and cannot be replaced. We did, however, replace some references that we deemed appropriate to newer ones and also included more recent publications when applicable. References 16 (L536), 17 (L539), 35 (L588), 36 (L592), 51 (L644), 54 (L652) were replaced by newer ones and two additional references were included (L 669; L671). As suggested, a more clinical application was included in the conclusion (L489).

Reviewer #2: General comments.

Methionine and cysteine oxidation was measured at various levels of dietary doses Cys in neonatal pigs to quantify “Cys-sparing effects” on Met oxidation. Inferences based on enzymatic assays and animal growth responses have estimated Cys-sparing effects in animals, however this is the first direct report to quantify Cys sparing effects based on changes in Met oxidation in neonatal pigs (or other animals to my knowledge). The methods and experimental design and techniques are well described and sufficient to answer the questions proposed. As noted in the following specific comments, alternate statistical analysis may provide more quantitative values, although the overall inferences are not likely to be altered. Using 1-14C Met and 1-14C Cys, strong data are reported to support that the Cys-sparing effect occurs by a dose-response inhibition of Met oxidation through the transsulfuration pathway.

The manuscript and research effort is quite refreshing and contributes valuable information on sulfur amino acid metabolism.

Dear Reviewer, 

Thank you for your thoughtful comments. 

In response to your statistics questions, we did consider this and explore these possibilities; however, as you point out they did are not likely to affect the overall inferences and final outcomes as the statistical analysis used have been widely applied in the literature. The model assumptions for the model used to analyzed plasma AA concentrations were checked for each variable of interest, and when necessary, data was transform to meet all the assumptions. Thus, we do not believe that there is a “lack of fit” of the data. As such, the data was not reanalyzed. Responses to specific comments are provided below.

Specific Comments.

L 27. Should be, “Methionine (Met) is an…..”

Thank you for catching that. 

L 75 (and throughout the text). Sentences should be restated to avoid use of phrases “It has been demonstrated”; or “it was found that” (L 79, L 81, L 414, L 416, L 422

These sentences were restated as suggested.

L 93 Would the lack of a reduction in Met oxidation in the presence of excess Cys infer a coupling of Cys synthesis to Met oxidation – ie., does a preferential conversion of homocysteine to Cys occur even if Cys is in excess.

That would be unlikely in a case where animals are deficient in TSAA, as the priority would be to direct Met for protein synthesis (reviewed by Ball et al., 2006); and this is likely even greater in our current study where the animals were growing at high rates. There may be metabolic conditions in which there is a higher demand for secondary metabolites from the transsulfuration pathway (e.g., glutathione) that would increase the flux of the transsulfuration pathway resulting in a higher requirement for the TSAA. We don’t believe that this was the case in our study as we observed a reduction in Met oxidation until the requirement for the TSAA was met. 

L 120. Correct "Piglets weighed" not "weighted"

Great catch. Corrected.

L 195. Responses fitted to a non-linear model would provide greater statistical robustness that the multiple comparison approach used, although the over inferences may not change.

We agree that using a non-liner model may provide greater statistical robustness. However, we used ANOVA and a two-way linear crossover model to evaluate the effects of dietary intake of Cys on Met oxidation. If only ANOVA was applied to investigate this effect, we agree that the statistical analysis should be redone. However, that’s not the case as this analysis is just complementary to the two-way linear crossover model, which is still widely used (Ennis et al., 2020; Martin et al., 2019; Packer et al., 2017). Moreover, the over inferences will likely not change as stated.

L 202. Were the iterative partitions subjective or quantitative using a linear-plateau model?

For all two-way linear cross over models, we assessed by each diet level in an iterative fashion and the partition that resulted in the best fit, as assessed by the AIC, BIC, and R, were chosen. 

L 297. Responses seem to be a “lack of fit”. A more rigorous statistical analysis is needed for these inferences. Were variances normally distributed across the range of Cys inputs? This concern also affects inferences on Tau and Thr concentrations. Are the quadratic responses being “detected” because of a greater variance on results ate the extremes of Cys intakes, ie., (L 310 to 311). Also L 446 to 449).

We do not believe that the responses are a “lack of fit”. We used the appropriate statistical model for the experimental design. The assumptions for generalized linear mixed models were evaluated, which include homoscedasticity of residuals, and when violated, modification in the covariance structure and/or log transformation were applied. Thus, we don’t believe that there is a lack of fit as we met the model assumptions. Polynomial contrasts can be applied in this data set and we have sufficient number of treatments to do so. Our statistical model was further reviewed and approved by a statistician at the University of Guelph. 

L 387 to 389. Perhaps another explanation relates to the statistical models resulting in a “lack of fit”, transformation of the data to adjust for non-normal distributions of variances may eliminate the responses that are somewhat difficult to explain.

Again, the assumptions for generalized linear mixed models were evaluated, which include homoscedasticity of residuals, and when violated, (modification in the covariance structure and/or log transformation were applied in the dataset). Thus, we don’t believe that there is a lack of fit as we met the model assumptions. Plasma amino acids are static measurements and do not provide a deep understanding of the in vivo kinetics, as we discussed in the text (unless they are evaluated under a meal response). The plasma amino acid response is complementary to our isotope dilution technique, which provides a more in-depth evaluation of the dynamic changes occurring in the body through measurements of metabolic fluxes. 

Dear Dr. Shoveller,

Thank you for submitting your manuscript to PLOS ONE. After careful consideration, we feel that it has merit but does not fully meet PLOS ONE’s publication criteria as it currently stands. Therefore, we invite you to submit a revised version of the manuscript that addresses the points raised during the review process.

- please do follow directions provided by reviewers to improve your manuscript

- discuss the relationship of hypermethionine diet on brain oxidative stress and consecutive behavioral changes 

- clearly emphasize the limitations of the study in the discussion section 

We look forward to receiving your revised manuscript.

Kind regards,

Dragan Hrncic

Academic Editor

Dear Dr. Hrncic, 

Thank you for your suggestions. We addressed the effects of hypermethionine on brain oxidative stress and consecutive behavioral changes (L477) and also the limitations of our study mostly related to the plasma AA data (L465). Further, we tried to address all comments provided by the reviewers and we provided reasoning if minor suggestions were not substantially incorporated in the manuscript. 

This study was conducted in 2002-2003 and was approved by Faculty of Agriculture, Forestry and Home Economics Animal Policy and Welfare Committee at the University of Alberta. The University Animal Care Committee could not provide the AUP number because it has been done 20 years ago. 

We hope the modifications are deemed appropriate and that the Journal Editorial staff considers the manuscript to be a meaningful contribution to the body of knowledge in sulfur amino acids metabolism and animal biology.

---

## [Decision Letter · Decision Letter 1]

22 Sep 2022

Methionine and cysteine oxidation are regulated in a dose dependent manner by dietary Cys intake in neonatal piglets receiving enteral nutrition

PONE-D-22-13606R1

Dear Dr. Shoveller,

We’re pleased to inform you that your manuscript has been judged scientifically suitable for publication and will be formally accepted for publication once it meets all outstanding technical requirements.

Kind regards,

Dragan Hrncic

Academic Editor

PLOS ONE

Additional Editor Comments (optional):

Reviewers' comments:

Reviewer's Responses to Questions

**Comments to the Author**

1. If the authors have adequately addressed your comments raised in a previous round of review and you feel that this manuscript is now acceptable for publication, you may indicate that here to bypass the “Comments to the Author” section, enter your conflict of interest statement in the “Confidential to Editor” section, and submit your "Accept" recommendation.

Reviewer #2: All comments have been addressed

2. Is the manuscript technically sound, and do the data support the conclusions?

Reviewer #2: Yes

3. Has the statistical analysis been performed appropriately and rigorously? 

Reviewer #2: Yes

4. Have the authors made all data underlying the findings in their manuscript fully available?

Reviewer #2: Yes

5. Is the manuscript presented in an intelligible fashion and written in standard English?

Reviewer #2: Yes

6. Review Comments to the Author

Reviewer #2: Acceptable responses or modifications have been included in the revision.

Some very minor edits are suggested before the galley proofs are released.

L 195 In the equation 2 denominator - isotope is misspelled.

L 201 change "thle" to "the"

L 329.... was observed "for".

L 467. Suggest, Due to the invasive procedures of.....

7. PLOS authors have the option to publish the peer review history of their article (what does this mean?). If published, this will include your full peer review and any attached files.

Reviewer #2: **Yes: **Thomas D. Crenshaw

---

## [Editor Report · Acceptance letter]

28 Sep 2022

PONE-D-22-13606R1 

Methionine and cysteine oxidation are regulated in a dose dependent manner by dietary Cys intake in neonatal piglets receiving enteral nutrition 

Dear Dr. Shoveller:

I'm pleased to inform you that your manuscript has been deemed suitable for publication in PLOS ONE. Congratulations! Your manuscript is now with our production department. 

Kind regards, 

on behalf of

Professor Dragan Hrncic 

Academic Editor

PLOS ONE